# Vocal Emotion Perception and Musicality—Insights from EEG Decoding

**DOI:** 10.3390/s25061669

**Published:** 2025-03-08

**Authors:** Johannes M. Lehnen, Stefan R. Schweinberger, Christine Nussbaum

**Affiliations:** 1Department of Clinical Psychology in Childhood and Adolescence, Friedrich Schiller University Jena, 07743 Jena, Germany; 2Department for General Psychology and Cognitive Neuroscience, Friedrich Schiller University Jena, 07743 Jena, Germany; christine.nussbaum@uni-jena.de; 3Voice Research Unit, Friedrich Schiller University Jena, 07743 Jena, Germany; 4Swiss Center for Affective Sciences, University of Geneva, 1205 Geneva, Switzerland

**Keywords:** vocal emotion perception, musicality, fundamental frequency (F0), timbre, EEG, neural decoding

## Abstract

Musicians have an advantage in recognizing vocal emotions compared to non-musicians, a performance advantage often attributed to enhanced early auditory sensitivity to pitch. Yet a previous ERP study only detected group differences from 500 ms onward, suggesting that conventional ERP analyses might not be sensitive enough to detect early neural effects. To address this, we re-analyzed EEG data from 38 musicians and 39 non-musicians engaged in a vocal emotion perception task. Stimuli were generated using parameter-specific voice morphing to preserve emotional cues in either the pitch contour (F0) or timbre. By employing a neural decoding framework with a Linear Discriminant Analysis classifier, we tracked the evolution of emotion representations over time in the EEG signal. Converging with the previous ERP study, our findings reveal that musicians—but not non-musicians—exhibited significant emotion decoding between 500 and 900 ms after stimulus onset, a pattern observed for F0-Morphs only. These results suggest that musicians’ superior vocal emotion recognition arises from more effective integration of pitch information during later processing stages rather than from enhanced early sensory encoding. Our study also demonstrates the potential of neural decoding approaches using EEG brain activity as a biological sensor for unraveling the temporal dynamics of voice perception.

## 1. Introduction

Attending to the emotional connotations expressed in human voices is something we do on a daily basis. Whether an angry outburst in busy traffic, the gentle expression of affection by a loved one, or the scream of an unfortunate protagonist in our favorite horror movie, emotional vocalizations are all around us. They form an essential basis for social interactions, providing information about the emotional states of others, and are vital for adequate and prompt responses such as empathy, caution, or even fight-or-flight responses [1].

In an effort to increase our understanding of vocal emotion processing, researchers used electroencephalography (EEG) to investigate the corresponding neuro-temporal dynamics in the brain. This cumulative work revealed a discrete time-course comprising a series of hierarchical sub-processes, which may be linked to distinct electrophysiological components [1,2] or time windows of neural processing [3]. The first stage consists of basic sensory processing of the auditory input and is assumed to be reflected in a negative ERP component emerging shortly after stimulus onset, the N100 [1,4]. In the second stage, these cues are evaluated for emotional meaning and integrated with further emotional processing, which has been thought to manifest in modulations of subsequent responses, including the P200 component [5] and the mismatch negativity (MMN) [6]. Finally, in the third stage, this information is relayed to higher-order cognition, such as semantic processing or evaluative judgements like emotion appraisal [7], and to response preparation. In the EEG time-course, these processes have been linked to the N400 component [8], the late positive potential (LPP) [5], and the (lateralized) readiness potential [9], respectively.

Discernable in the first stage, a crucial aspect of vocal emotion perception lies in the processing of acoustic markers, such as the fundamental frequency (F0) and timbre (put simply, any perceivable difference in voice quality between two voices that have identical F0, intensity, and temporal structure), and how their specific profiles vary in different emotions [2,10,11,12]. However, how the human brain precisely extracts emotional meaning from the acoustic input remains insufficiently understood [2]. Nussbaum et al. [13] targeted this research gap by exploring how the processing of F0 and timbre links to the aforementioned ERP components. They employed a parameter-specific voice morphing approach to create stimuli which contain emotional information in F0 or timbre only, while other parameters were kept emotionally non-informative. Their experiment not only revealed behavioral effects, but also modulations of the P200 and N400 by these vocal cues, reflecting their relative importance for discerning specific emotions: while F0 was found to be relatively more important for the perception of happiness, sadness, and fear in voices, timbre was found to be more relevant for the perception of pleasure.

A notable phenomenon described in the literature regards the profound inter-individual differences in vocal emotion perception [2,8,14,15]. A relevant trait in this context is musicality [16]. The interconnection between musical expertise and vocal emotion perception has been shown on a behavioral level, with musicians being significantly more accurate in discriminating between vocal emotions than non-musicians [16,17,18,19]. A recent behavioral study suggests that this advantage is based on increased auditory sensitivity, specifically towards the pitch contour of the voice, characterized by the fundamental frequency (F0), rather than towards timbre [20]. It is, however, impossible to discern the functional and neural locus of this increased sensitivity from the behavioral response alone. Specifically, the degree to which differences linked to musicality originate from early representations of acoustic information or from later, more controlled aspects of emotional processing and decision making remains unclear.

To address this, Nussbaum et al. [21] recorded the EEG of substantial samples of musicians and non-musicians. As in their previous study [13], Nussbaum et al. presented vocal stimuli manipulated with parameter-specific voice morphing, with emotional content expressed in F0 or timbre only. They found that the P200 and even later N400 component did not differ significantly between groups regardless of any acoustic manipulation. Instead, group differences were observed for a later (>500 ms), central–parietal LPP. While non-musicians displayed similar reductions in the LPP in response to acoustic manipulation, musicians showed an emotion-specific pattern: For negative emotions, such as fear and sadness, musicians’ LPP was unaffected by the acoustic manipulation, suggesting that musicians were well able to compensate for a missing parameter. For happiness and pleasure, however, the LPP was reduced to a larger extent when F0 (as compared to timbre) was rendered uninformative, potentially supporting the special reliance on F0 cues reported in the behavioral data. The authors conceded that the observation of differences in later time windows only was unexpected and discussed potential explanations. On the one hand, this finding could indicate that musicians’ higher auditory sensitivity is reflected in the more efficient higher-order integration of acoustic information rather than a facilitation of early bottom-up analysis processes. On the other hand, earlier (<500 ms) group differences in neural processing cannot be ruled out from these data because scalp-recorded ERPs can be relatively blind to a subset of combinations of deep neural generators, because a conventional ERP analysis might simply lack the statistical power needed to uncover these effects in their data, or both.

Consequently, Nussbaum et al. [21] suggest using additional analysis methods, such as machine learning-based multivariate pattern analyses (MVPAs). MVPA is a computational method used to analyze complex neural data by identifying patterns of brain activity across multiple variables, such as channels or timepoints in the EEG [22], or voxels in brain imaging data [23]. Unlike univariate approaches, which focus on individual brain regions or signals, MVPA evaluates the distributed nature of neural representations, making it particularly useful for decoding cognitive states, predicting behaviors, or tracing sensory perception [24]. For instance, in recent years, MVPA has gained popularity in the field of visual cognition (e.g., [25,26,27]), where the goal is to understand how patterns of neural activity correspond to specific perceptual states or processes. MVPA has been shown to have an increased sensitivity towards small effects [22,28]. In addition, it allows for better consideration of the continuous nature of cognitive processes as the EEG data are analyzed in an unconstrained manner [29] in contrast to a component-focused conventional ERP analysis [22,30]. There are indeed examples where an additional MVPA could uncover early EEG effects, which a conventional ERP analysis failed to discover (e.g., [31]).

To this end, the aim of the present study was to re-analyze the data obtained by Nussbaum et al. [21] using a time-series multivariate decoding approach. We used their EEG dataset of musicians and non-musicians on vocal emotion perception to train a linear discriminant classifier (LDA) to discriminate between the four presented emotional categories (happiness, sadness, fear, and pleasure). This approach allows for the representation of emotion to be finely traced throughout the processing time-course with the goal of providing a more detailed picture of the relationship between musicality and vocal emotion processing. Based on the findings obtained by Nussbaum et al. [21], we expected significant decoding in later parts of the epoch (>500 ms) but were also interested in whether this exploratory approach would reveal additional early decoding effects (<500 ms).

## 2. Materials and Methods

This study presents a reanalysis of the data collected for Nussbaum et al. [21] with an EEG decoding approach to complement their ERP analysis. Thus, many of the methodological aspects overlap, and we refer to their study where appropriate. In essence, while the reported participants, stimuli, design, and recording procedure remain unchanged, the data preprocessing method, analysis approach, and statistical testing method constitute the novel aspects of the present methodology.

### 2.1. Participants

We analyzed data from 38 (semi-) professional musicians (17 males and 21 females, aged 20 to 42 years [M = 30; SD = 5.54]) and 39 non-musicians (19 males and 20 females, aged 19 to 48 years [M = 30.5; SD = 6.34]). One musician had to be excluded from analysis, due to technical difficulties during the recording process, meaning a final sample of 77 participants entered analysis. Musicianship was defined as either having an academic music-related degree or having non-academic music qualifications (i.e., working as a musician or winning a music competition) in conjunction with extensive musical training. In this sample, thirty-five participants had studied their instruments for over 10 years (three had studied between 6 and 9 years, and one between 4 and 5 years). While professional musicians pursued music as their main career, semi-professional musicians held other jobs but still dedicated a majority of their time to playing music. Non-musicians played no instrument and did not engage in musical activities during childhood, with a few exceptions receiving mandatory flute training in primary school. For further details on the participants, see [20] and its supplementary material. All participants were fluent in German and provided informed consent before completing the experiment. Data collection was pseudonymized, and participants were compensated with EUR 25 or with course credit. The experiment was in line with the ethical guidelines of the German Society of Psychology (DGPs) and approved by the local ethics committee (Reg.-Nr. FSV 19/045).

### 2.2. Stimuli

Ninety-six original audio recordings were used for morphing, comprising eight speakers (four male and four female) uttering three pseudowords (/molen/, /loman/, /belam/) with expressions of happiness, pleasure, fear, and sadness. Voice morphs were created using the Tandem-STRAIGHT software (https://www.isc.meiji.ac.jp/~mmorise/straight/english/introductions.html, accessed on 3 March 2025) [32] on the trajectory between emotional utterances and emotional averages for each speaker and pseudoword (Figure 1). Emotional averages were interpolations of all four emotions and, while not necessarily fully neutral, were emotionally uninformative with respect to the present task structure. They were chosen over neutral reference voices because a previous study found them to sound more natural when used in conjunction with parameter-specific voice morphing [33]. Full-Morphs were stimuli with all parameters taken from the emotional version (corresponding to 100% from the specific emotion and 0% from the average), except for the timing parameter, which was taken from the emotional average (corresponding to 0% specific emotion and 100% emotional average). Here, timing refers to the sequential progression of vocal sounds over time and is realized via manually assigned time anchor positions [34]. F0-Morphs were stimuli with the F0-contour taken from the specific emotion, but timbre and timing were taken from the emotional average. Conversely, Timbre-Morphs were stimuli with all timbre parameters (i.e., formant frequency, spectral level, and aperiodicity) taken from the specific emotion except for F0 and timing, which were taken from the emotional average. We also included the emotional averages for exploratory purposes. In total, this resulted in the following: 8 (speakers) × 3 (pseudowords) × 4 (emotions) × 3 (morphing conditions) + 24 average (8 speakers × 3 pseudowords) = 312 stimuli. Using PRAAT [35], we normalized all stimuli to a root mean square of 70 dB SPL (duration M = 780 ms, range = 620–967 ms, SD = 98 ms). Example stimuli can be found on the OSF repository associated with the original publication by Nussbaum et al. [21] at https://osf.io/6vjh5/ (accessed on 3 March 2025).

### 2.3. Design

#### 2.3.1. EEG-Setup

The EEG was recorded using a 64-channel BioSemi Active II system (BioSemi, Amsterdam, The Netherlands) with electrodes attached to a cap in accordance with the extended 10–20 system (electrode specifications as in [21]). The horizontal electrooculogram (EOG) was recorded from two electrodes at the outer canthi of both eyes, and the vertical EOG was monitored with a pair of electrodes attached above and below the right eye. Data quality was ensured for each participant and electrode during the preparation for the EEG recording. Electrode offset was kept in the range of ± 40 mV. Before data collection started, it was checked that all electrodes measured a clean signal. Signal quality was then monitored by the experimenter during the whole experiment. The sound stimuli were presented via in-ear headphones (Bose^®^MIE2 mobile headset, Bose, Framingham, MA, USA). During recording, participants were seated in a dimly lit, electrically shielded, and sound-attenuated chamber (400-A-CT-Special, Industrial Acoustics^TM^, Niederkrüchten, Germany) with their head seated on a chin rest 90 cm from the computer screen. For the presentation of the written instructions and the stimuli, we used E-Prime 3.0 [36].

#### 2.3.2. Procedure

Participants were instructed to attentively listen to vocal stimuli and identify the emotions conveyed. In the beginning of each trial, a white fixation cross was displayed on a black background. After a randomly jittered interval of 1000 ± 100 ms, the fixation cross turned green, and a vocal stimulus was presented. This was followed by a 2000 ms period of silence during which the green fixation cross remained visible. In 10% of the trials, a screen presenting four response options appeared immediately after the stimulus ended and stayed on until a response was made. This unpredictable prompt was designed both to sustain focus on the emotional content and to minimize motor response confounds. Accordingly, these response trials were retained in the analysis since participants were prompted after voice offset and could not predict whether a response would be required due to the randomized prompting. Notably, behavioral response accuracy was in line with previous findings [13,20] (see supplementary material in [21]). The participants had the opportunity to practice the task with 20 initial practice trials, which consisted of stimuli not employed in the main experiment. Next followed 312 experimental trials presented in random order and then repeated in a different random order; thus, the whole experiment consisted of 624 trials. The trials were sectioned into eight blocks, where each block contained 78 trials and a break between each block. The whole experiment lasted approximately 60 min.

### 2.4. EEG Preprocessing

While based on Nussbaum et al. [21], the preprocessing of the EEG data was redesigned and then re-conducted to align with the new analysis approach. It was performed using EEGLAB v2023.0 [37] within MATLAB 2018b [38]. Data were re-sampled to 250 Hz and re-referenced to the average reference. The EEG was then sequenced into individual trial epochs from −200 to 1000 ms relative to stimulus onset, and the four eye channels were removed for further data analysis. The data were baseline-corrected by subtracting the average pre-stimulus signal [−200 to 0 ms] and further down-sampled to 100 Hz to increase the signal-to-noise ratio by averaging timepoints together [22]. For down-sampling, we used FieldTrip v20220429 [39] and its “ft_resampledata” function, which detrends the EEG to prevent aliasing. No filtering was applied to the EEG signal to avoid distortions and subsequent artifacts in the decoding process following the recommendations of previous studies [22,40,41]. Furthermore, no additional artifact or channel rejection was performed as the classifier can learn to suppress noisy epochs and channels during the training process [22].

### 2.5. Vocal Emotion Decoding

The four vocal emotions (happiness, pleasure, sadness, and fear) were classified from EEG activity using the CoSMoMVPA toolbox (https://www.cosmomvpa.org/) [42] and their linear discriminant classifier (LDA). Training and testing were performed in a leave-one-trial-out cross-validation design in which the algorithm is trained on all but one trial, which is then used for testing. This is repeated until every trial has served as a testing trial. Classifier performance is then averaged across these repetitions (Figure 2B; see [27,43,44] for example studies using this approach). Classification was performed separately for each participant and on all available electrodes. Classification moved in a time-resolved manner, where each 10 ms time window of the 1000 ms epoch was analyzed individually (Figure 2C). The resulting accuracy-over-time distributions were smoothed by averaging five timepoints into one in a rolling-mean approach [27].

### 2.6. Training and Testing Procedure

Emotion decoding of the EEG data was performed in three separate training and testing schemes. First, the classifier was trained and tested on all available stimuli. This was performed to obtain insights into the general temporal pattern of vocal emotion decoding and to achieve most robust classification by using all available data. In two follow-up analyses, the classifier was trained on data from Full-Morphs but only tested on data from either F0- or Timbre-Morphs to investigate the importance of these parameters for processing vocal emotions. The aim was to obtain a system which is trained to classify normal full-spectrum stimuli, comparable to a human brain which is trained to perceive natural voices, and then investigate how classification performance is affected by missing F0 or timbre information in the brain activity of the test trials.

### 2.7. Statistical Testing

Decoding accuracies were tested against chance level using threshold-free cluster enhancement in conjunction with a one-tailed permutation test over 10,000 iterations [45]. To investigate how emotions are processed in our sample regardless of musicality, we first tested the average classification performance *across* groups against the bootstrapped distribution obtained in the permutation test. Then, to infer how decoding is affected by musicality, we separately tested the average classification performance within each group. The *p* values were thresholded with an alpha level of 0.05 and corrected for multiple comparisons using the Monte Carlo method.

## 3. Results

### 3.1. General Emotion Decoding

First, we decoded vocal emotion across all recorded participants regardless of musicality level. This was performed as a proof of concept and to compare the results to the previous literature. When training and testing on all stimuli, significant decoding emerged in a small elusive peak at around 300 ms (Figure 3, top row, left: peak accuracy = 25.4%, *p* = 0.049) and followed by continuous decoding between 420 and 760 ms (peak accuracy = 25.9%, *p* < 0.001). Finally, a very late peak emerged from 870 ms until the end of the analyzed epoch (peak accuracy = 25.8%, *p* = 0.004). Following this analysis, we investigated the role of F0 on vocal emotion processing by training the classifier on the Full-Morphs and testing it with F0-Morphs only. Here, significant emotion decoding was observed at around 450 ms (Figure 3, middle row, left: peak accuracy = 25.7%, *p* = 0.037) and between 540 and 630 ms (peak accuracy = 25.9%, *p* < 0.001). Finally, we investigated the role of timbre in the same manner: by training on Full-Morphs and testing on Timbre-Morphs only. However, there was no significant emotion decoding in this analysis (Figure 3, bottom row, left).

### 3.2. Emotion Decoding and Musicality

Having looked at vocal emotion classification patterns across all participants, we turned to the role of musicality by training and testing the classifier within the individual participant groups. The training and testing schemes were identical to the analysis across groups. First, training and testing on all stimuli revealed significant above-chance emotion decoding for musicians. Two peaks emerged, one between 500 and 700 ms after stimulus onset (Figure 3, top row, right: peak accuracy = 26.3%, *p* < 0.001) and a later peak from 870 ms to the end of the epoch (peak accuracy = 26.1%, *p* = 0.007). When classifying data from non-musicians, a small peak of two timepoints emerged at 430 ms after stimulus onset (peak accuracy = 25.6%, *p* = 0.044). When testing on F0-Morphs only, significant decoding emerged for musicians in two peaks from 550 to 630 ms (Figure 3, middle row, right: peak accuracy = 26.2%, *p* = 0.004) and from 880 to 930 ms after stimulus onset (peak accuracy = 26.2%, *p* = 0.002). By contrast, there was no significant decoding for the classification of data from non-musicians. Finally, when training on all stimuli but only testing on Timbre-Morphs, no significant decoding was revealed in either group (Figure 3, bottom row, right).

## 4. Discussion

The neural correlates of individual differences in emotion recognition ability are a pivotal area of ongoing research [46]. The aim of this study was to investigate the neural differences in vocal emotion perception of musicians and non-musicians, considering that musicality could be one of the determinants of individual differences in vocal emotion perception [20]. To this end, we performed a re-analysis of EEG data first reported by Nussbaum et al. [21] with a neural decoding approach. Together with this previous paper, a distinctive feature of the present study is that it allows a direct comparison of neural decoding and ERP data from the same experiment.

Overall, we succeeded in decoding representations of vocal emotions from the EEG. We observed significant and robust decoding beginning at 400 ms after stimulus onset for decoding on all trials and across groups. When comparing musicians with non-musicians, there was significant emotion decoding on data from musicians starting from 500 ms to the end of the epoch, which contrasted with very few significant timepoints at best on the data from non-musicians. Furthermore, when looking at classification based on F0-morphed stimuli, the same pattern persists, with significant decoding emerging for musicians but not for non-musicians. Thus, for Full- and F0-Morphs, it seems that the emotions participants listened to can be somewhat extracted from the musicians’ EEG responses, but not from the responses of non-musicians. For the classification of timbre-morphed stimuli, however, no substantial EEG decoding of vocal emotions could be discerned in either group. Most importantly—and in line with the previous ERP analysis—there were no significant effects before 500 ms, even with a more sensitive decoding analysis.

In general, these results suggest that the EEG of musicians contains more traceable information about the emotional categories than that of non-musicians [24] and that this mainly manifests in later stages of the processing stream. This converges with the previous ERP findings, which only discovered group differences in the LPP component between 500 and 1000 ms and not earlier. In addition, our results also highlight the importance of F0 as most of the emotional information in the musicians’ EEG seems to be contained in the fundamental frequency. This is in line with the previous literature [47,48], specifically recent behavioral evidence, where musicians exhibited an advantage in perceiving emotions expressed via F0 only [20] compared to emotions expressed via timbre. Within that study, the behavioral classification of Timbre-Morphs was also closest to the guessing rate, which could explain why no decoding emerged for Timbre-Morphs in our analysis.

As discussed by Nussbaum et al. [21], it may seem surprising that group differences appeared no earlier than 500 ms, as one could have expected to find differences in early acoustic-driven processes. However, the EEG data suggest that the differences in the auditory sensitivity towards emotions are primarily reflected in later, more top-down-regulated complex processes such as emotional appraisal [2,7]. One may speculate that musicians do not differ from non-musicians in the extraction of F0 information during early auditory processing but may be more efficient in using the extracted F0 information for emotional evaluation and conscious decision making. Thus, the absence of early differences between groups is not a result of a lack in statistical power but reveals that musicality indeed shapes later stages of emotional processing. In the same vein, one might be tempted to argue that the timing effects of the stimuli could have influenced the present results. For instance, a late decoding effect for musicians could either mean that neural decoding indeed took considerable time to occur, that emotional information became available only later in the stimuli and triggered an early brain response, which was therefore visible in later EEG activity, or anything in between these extreme alternatives. However, the fact that other work found early ERP differences between emotions with the present stimuli [13] is more consistent with the present interpretation that, even though emotional information is available early in the stimuli, neural decoding for musicians emerges in later processing stages. Additionally, potential confounds related to varying stimulus durations and offsets can be ruled out as the timing parameter was consistently taken from the emotional average during the morphing process (Figure 1). While individual stimuli may have varied slightly in the duration between pseudowords and speakers, the timing parameter was kept consistent across emotions.

This study presents a remarkable convergence of findings across different analyses of EEG data. In both the ERP and the decoding approach, differences were found >500 ms. However, in the ERP analysis, group effects were only observed for some of the emotions, namely happiness and pleasure. The present decoding approach provides a more holistic picture as it returns classification success across all emotions. Thus, our findings highlight the added value of analyzing the same EEG data using different methods as they help us understand how the idiosyncrasies of different methods shape the inferences we make from the data.

Despite these intriguing findings, we must consider several limitations of this study. Even when statistically robust, the observed above-chance decoding accuracies are low, suggesting that the observed effects are relatively small. This could be related to several potential reasons: First, as this study presents an exploratory re-analysis, it was not originally designed for a neural decoding analysis. As a consequence, the number of trials is not ideal and lower than that in studies with similar approaches from the visual domain (e.g., [43,49,50,51]), which will likely have impacted classification performance. Second, we used a linear classification algorithm to decode the emotional representations, which can have lower performance than non-linear deep learning approaches, for example [52]. As deep learning algorithms require large amounts of training data for robust classification [53], we considered a more traditional machine learning application to be the better choice for our dataset. Even so, for neuroscientific decoding studies interested in the representations of cognition and not prediction, such as in brain–computer-interface (BCI) applications [54,55], the focus lies more on the fact that information patterns are indeed *present* in the data. With this in mind, even low decoding accuracies provide useful insights if they are statistically robust and thus generalize across the population [24].

Finally, the question might arise whether the modulatory role of musicality on vocal emotion perception could be further qualified by individual differences in hormonal status, especially in women during the menstrual cycle. There is some evidence that people adapt their voice depending on vocal characteristics of an interaction partner and that these changes are modulated by the menstrual cycle phase in women [56]. It has also been proposed that some musical skills might reflect qualities that are relevant in the context of mate choice, and thus, that women might be expected to show an enhanced preference for acoustically complex music when they are most fertile. Unfortunately, this hypothesis was not confirmed upon testing [57]. Although future research in the field of the present study might delve into a possible role of more fine-grained aspects of individual differences (such as hormonal status), generating specific hypotheses about how hormonal influences could qualify the present results seems difficult at present.

Consequently, despite these limitations, these results give valuable insights into how vocal emotions are represented in the brain activities of musicians and non-musicians. They also highlight the general potential of neural decoding approaches in researching nonverbal aspects of voice perception and emotion perception from the human voice in particular. In recent years, these methods have become more and more popular with works in the visual domain, but only a few studies have employed EEG decoding to investigate auditory perception. These have mainly focused on decoding speech content [58,59] but less on how the brain processes auditory nonverbal information about speakers, such as emotional states. Giordano et al. [60] studied nonverbal emotional exclamations by decoding emotional information from time-series brain activity related to synthesized vocal bursts. In a recent study, musical and auditory emotions were identified from the EEG activity of cochlear implant users [61]; however, their focus was more on reaching high prediction accuracies from single-electrode recordings rather than tracking the time-specific correlates of the perception process itself. Thus, to our knowledge, this study is among the first to investigate the neuro-temporal correlates of individual differences in vocal emotion perception with a neural decoding approach with a direct comparison to standard ERP findings. While the advantages of MVPA over the traditional ERP method are increasingly recognized in visual perception research [29], its application in the auditory domain remains relatively underexplored. We anticipate that future studies within this field could greatly benefit from this method since the effects of musicality have been especially conflicting and hard to uncover [16,17,21]. The potential of the method to uncover the decoding of other acoustic parameters of emotions should also be further investigated, especially with regard to time-sensitive cues [3].

## 5. Conclusions

This study investigated the neuro-temporal dynamics of the advantage in vocal emotion perception provided by musicality using a multivariate pattern analysis to decode the neural representations of perceived vocal emotions from the brain activities of musicians and non-musicians. To study the degree to which vocal emotion perception relies on F0 or timbre cues, we further tested the classification of parameter-specific morphed stimuli containing emotional information within these parameters only. Significant and sustained decoding mainly emerged 500 ms after stimulus onset. Classification data for the individual groups suggest that this pattern was prominent for musicians but small or absent for non-musicians. Moreover, decoding in musicians appeared to be mainly a function of F0 information and was reduced to insignificance for Timbre-Morphs. Overall, this pattern confirms the interpretation that the advantage of musicality manifests in later, higher-order stages of auditory processing and relies on a superior integration of emotional information contained in the fundamental frequency of the voice. To the best of our knowledge, this study is among the first to investigate inter-individual differences in voice perception with machine learning methods. The results could highlight the potential for further applications of this approach in future studies in the field that focus on neural correlates of individual differences.

## Figures and Tables

**Figure 1 sensors-25-01669-f001:**
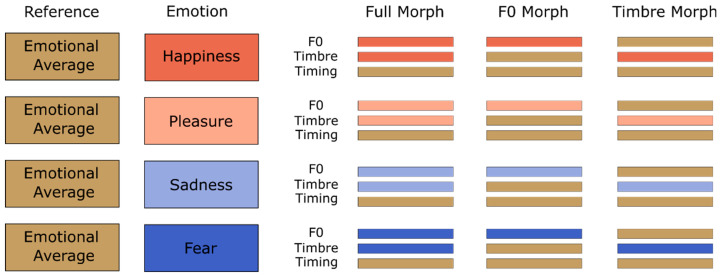
Morphing matrix for stimuli with averaged voices as reference (taken with permission from authors of [21]).

**Figure 2 sensors-25-01669-f002:**
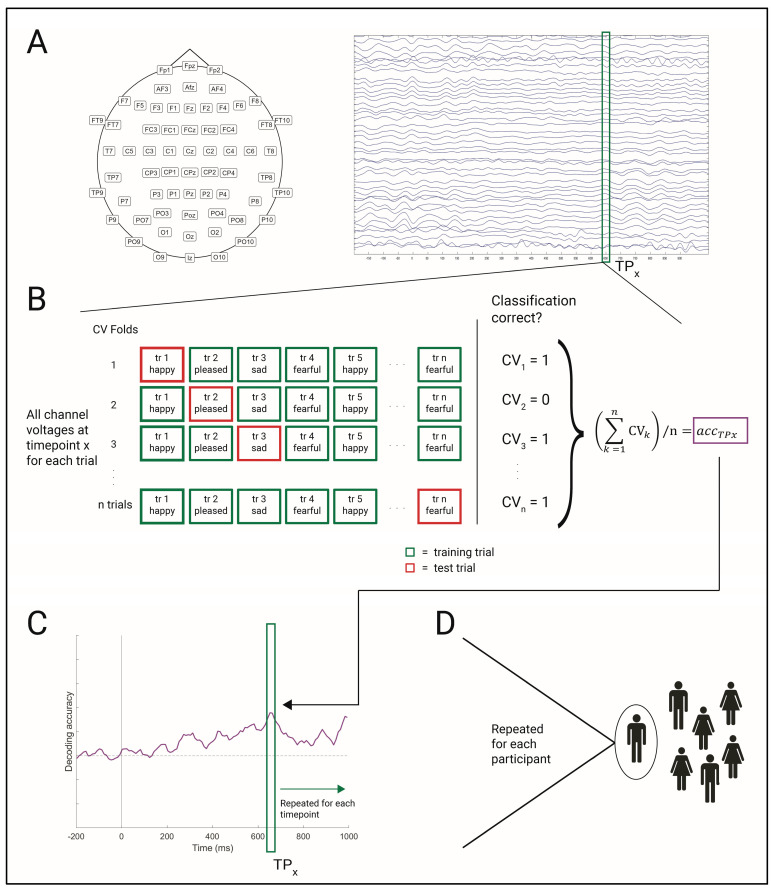
An illustration of the time-resolved EEG classification approach with a leave-one-out cross-validation design. (**A**) For all trials entering analysis, the channel voltages at a timepoint *x* (TPx) are extracted. (**B**) The trial information of timepoint *x* then enters classification analysis. First, trials are partitioned into cross-validation (CV) folds. In each fold, all trials but one are used to train the classifier, with the final trial being reserved for testing. Partitioning is repeated until each trial has served as a test trial once, resulting in *n* folds, with *n* being equal to the number of experimental trials. The algorithm is tested on the respective test trial within each fold, resulting in a correct (CV=1) or false (CV=0) prediction. The classification results of all CV folds are then averaged into an overall classification accuracy at timepoint *x* (${acc}_{TPx}$). (**C**) Steps A and B are repeated for each timepoint in the EEG epoch, resulting in an accuracy-over-time distribution. (**D**) This process is performed for each individual participant. In the final step, the accuracy distributions of all participants are averaged into an overall sample decoding distribution, which is then tested for statistical significance.

**Figure 3 sensors-25-01669-f003:**
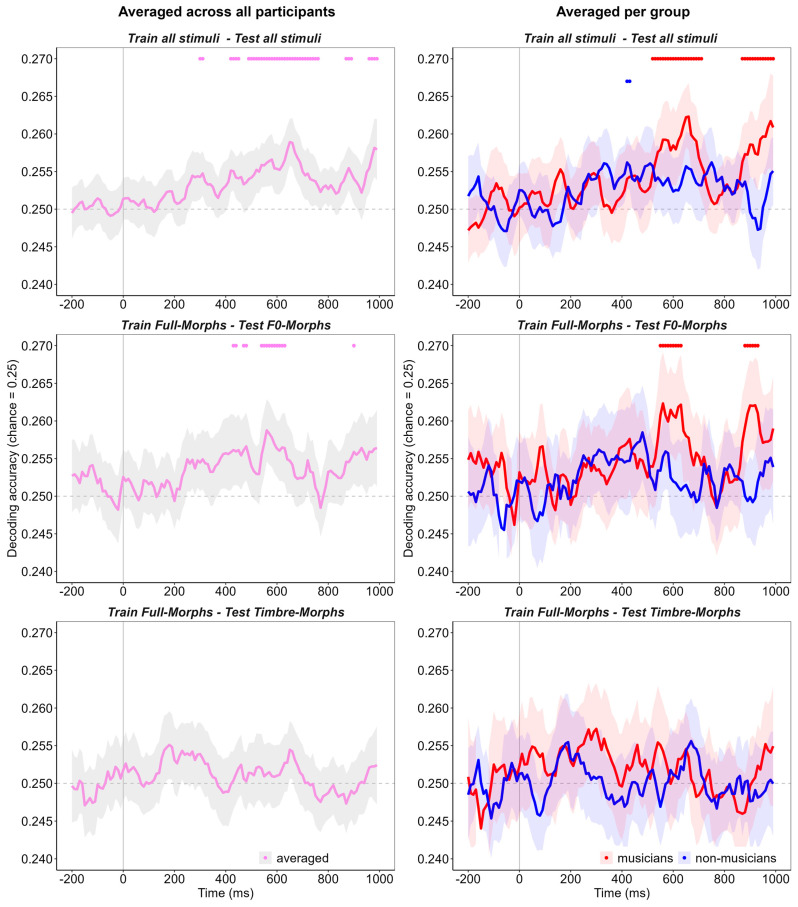
The results of the decoding analysis. The time continuum of the ERP epoch is denoted on the *X*-axis in milliseconds, with the pre-stimulus baseline interval beginning at −200 ms and stimulus onset beginning at 0 ms. The decoding accuracy is plotted on the *Y*-axis with the chance level of 0.25 signified by the dotted line. The left column shows the decoding results for decoding on all participants across groups. The right column shows decoding on musicians and controls, respectively. The dots above the accuracy curve mark significant timepoints, and the gray areas show the confidence intervals of decoding accuracy.

## Data Availability

The original data presented in this study as well as the analysis and visualization code are openly available in OSF at https://osf.io/qakne/ (accessed on 3 March 2025). Example stimuli can be found in OSF at https://osf.io/6vjh5/ (accessed on 3 March 2025).

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
