# Peer review of "Vocal Emotion Perception and Musicality—Insights from EEG Decoding"

_sensors, 2025, doi:10.3390/s25061669_

Round 1
Reviewer 1 Report
Comments and Suggestions for Authors
This study investigated the neuro-temporal dynamics of the advantage in vocal emotion perception provided by musicality, using a multivariate pattern analysis to decode the neural representations of perceived vocal emotions from the brain activity of musicians and non-musicians. This is an interesting study and demonstrates the potential of neural decoding methods. To make it even better, there are a couple of minor issues:
- In line 51 and 55, "mismatch negativity (MMN; 6)", " potential (LPP; 5)" . Here 6 and 5, are the citation of references? If yes, the citation is formatted incorrectly; if no, what it represents?
- Line 167, "timing parameter". By searching, in the field of music analysis, timing may refer to rhythm, note timing . May I ask what exactly is meant here and whether it needs to be specified.
- Line 228-230, No additional artifact or channel suppression is performed because the classifier can learn to suppress the noise Epoch and channel during training. Is the raw signal quality high enough? What is the value of offset current as a quality indicator in the experiment? Or are there other quality measures?
- In Fig.3, Considering contextual consistency, would it be better to standardize on "Full" instead of "All"? ("Train All - Test All")
Author Response
Comments 1: In line 51 and 55, "mismatch negativity (MMN; 6)", " potential (LPP; 5)" . Here 6 and 5, are the citation of references? If yes, the citation is formatted incorrectly; if no, what it represents?
Response 1: Thank you for pointing this out! This is indeed formatted incorrectly. We have changed it to the correct citation format. The changes can be found in the second paragraph of the introduction in line 51 and 55 respectively (marked in red.)
Comments 2: Line 167, "timing parameter". By searching, in the field of music analysis, timing may refer to rhythm, note timing . May I ask what exactly is meant here and whether it needs to be specified.
Response 2: We agree, for clarity timing in this context is missing a definition. Timing, in this case, refers to the acoustic timing of the vocal stimuli. We have included the following definition of timing, to make this more transparent: “Here, timing refers to the sequential progression of vocal sounds over time, and is realized via manually assigned time anchor positions (Kawahara & Skuk, 2018).” Please find it in section 2.2 – Stimuli in line 169 to 170 (marked in red).
Kawahara, H.; Skuk, V.G. Voice Morphing. In The Oxford Handbook of Voice Perception; Frühholz, S., Belin, P., Frühholz, S., Belin, P., Paulmann, S., Kotz, S.A., Eds.; Oxford University Press, 2018, ISBN 9780198743187.
Comments 3: Line 228-230, No additional artifact or channel suppression is performed because the classifier can learn to suppress the noise Epoch and channel during training. Is the raw signal quality high enough? What is the value of offset current as a quality indicator in the experiment? Or are there other quality measures?
Response 3: Thank you for pointing that out. We made sure that the raw EEG data was of sufficient quality during the data collection. We agree that more information on this would be beneficial. To clarify this further, we added the following section to the EEG-Setup paragraph of the methods section (line 192 – 195, marked in red):
“Data quality was ensured for each participant and electrode during the preparation for the EEG recording. Electrode offset was kept in the range of ± 40 mV. Before data collection started, it was checked that all electrodes measured a clean signal. Signal quality was then monitored by the experimenter during the whole experiment.“
Comments 4: In Fig.3, Considering contextual consistency, would it be better to standardize on "Full" instead of "All"? ("Train All - Test All")
Response 4: Thank you for your comment. We understand how the labeling of the figure, or the logic behind the training and testing procedure, could be misleading. Given the exploratory nature of our analysis, our primary goal was to establish a foundation and proof of concept—demonstrating that decoding vocal emotions from EEG is possible and that different groups exhibit distinct representations. To ensure the most robust decoding, our initial analysis included all available data from Full-Morphs, F0-Morphs, and Timbre-Morphs. Then, to investigate the specific roles of F0 and Timbre, we took an additional step. Since the human brain typically does not encounter vocalizations where emotional content is isolated to specific parameters (as in our F0- and Timbre-Morphs), we avoided giving the classifier an artificial “advantage” by training it on such data. Instead, to create a classification approach that better reflects how humans perceive voices, we trained the classifier only on Full-Morphs and then tested it separately on F0-Morphs and Timbre-Morphs.
To clarify this better, we have updated the labeling of Figure 3 to: 'Train all stimuli – Test all stimuli,' 'Train Full-Morphs – Test F0-Morphs,' and 'Train Full-Morphs – Test Timbre-Morphs.' These changes can be found in line 303 to 304 in the updated figure.
Reviewer 2 Report
Comments and Suggestions for Authors
The work presents interesting and expected data that the experience of musical activity should certainly influence the perception of emotional colouration of vocalists.
However, it would have been more convincing if the authors had presented data separately in a group of men and in at least two groups of women, divided by the phases of the menstrual cycle that are associated with fluctuations in sex steroid levels.
Estradiol and progesterone are known to modulate the balance of excitation and inhibition globally, so that evoked potentials, affective and even cognitive functions differ markedly during the menstrual cycle. It would be interesting to know whether this variation is related to the experience of musical activity.
Author Response
Comments 1: The work presents interesting and expected data that the experience of musical activity should certainly influence the perception of emotional colouration of vocalists.
However, it would have been more convincing if the authors had presented data separately in a group of men and in at least two groups of women, divided by the phases of the menstrual cycle that are associated with fluctuations in sex steroid levels.
Estradiol and progesterone are known to modulate the balance of excitation and inhibition globally, so that evoked potentials, affective and even cognitive functions differ markedly during the menstrual cycle. It would be interesting to know whether this variation is related to the experience of musical activity.
Response 1: This is an interesting possibility which we indeed had not considered when planning the present study. In general, we agree that hormonal changes during the menstrual cycle can systematically affect EEG, affective and cognitive functions, although this is rarely studied. To determine the relevance of this issue for the specific topic area of the present study, we decided to consult the literature. We performed various literature searches in Clarivate Web of Science, using the keywords “voice”, “music*”, “menstrual” and “cycle”. Unfortunately, we were unable to identify systematic sets of specific research findings that would have motivated an in-depth discussion of these hormonal factors in the context of the present study. That said, we did identify a few studies. Because we appreciate that the question could be of interest to some readers, we addressed the reviewer´s suggestion by adding a small portion of text in the discussion (limitation section line 409 – 420, marked in red), as follows.
“Finally, the question might arise whether the modulatory role of musicality on vocal emotion perception might be further qualified by individual differences in hormonal status, especially in women during the menstrual cycle. There is some evidence that people adapt their voice depending on vocal characteristics of an interaction partner, and that these changes are modulated by the menstrual cycle phase in women (Lobmaier et al., 2024). It also has been proposed that some musical skills might reflect qualities that are relevant in the context of mate choice, and thus that women might be expected to show enhanced preference for acoustically complex sounds when they are most fertile. Unfortunately, this hypothesis was not confirmed upon testing (Charlton et al., 2012). Although future research in the field of the present study might delve into a possible role of more fine-grained aspects of individual differences (such as hormonal status), specific hypotheses about how hormonal influences could qualify the present results seem difficult at present.”
Charlton, B. D., Filippi, P., & Fitch, W. T. (2012). Do Women Prefer More Complex Music around Ovulation? Plos One, 7(4), Article e35626. https://doi.org/10.1371/journal.pone.0035626
Lobmaier, J. S., Klatt, W. K., & Schweinberger, S. R. (2024). Voice of a woman: influence of interaction partner characteristics on cycle dependent vocal changes in women. Frontiers in Psychology, 15, 1401158-1401158. https://doi.org/10.3389/fpsyg.2024.1401158